# CD73: Friend or Foe in Lung Injury

**DOI:** 10.3390/ijms24065545

**Published:** 2023-03-14

**Authors:** Xiu-Min Hu, Nan-Rui Shi, Ji-Zhou Zhang, Yan-Qin Zuo, Xin Wang, Ya-Fei Zhao, Jia-Si Wu

**Affiliations:** School of Acupuncture and Tuina, Chengdu University of Traditional Medicine, Chengdu 610075, China

**Keywords:** CD73, purinergic receptor, lung injury, adenosine, inflammatory response

## Abstract

Ecto-5′-nucleotidase (CD73) plays a strategic role in calibrating the magnitude and chemical nature of purinergic signals that are delivered to immune cells. Its primary function is to convert extracellular ATP to adenosine in concert with ectonucleoside triphosphate diphosphohydrolase-1 (CD39) in normal tissues to limit an excessive immune response in many pathophysiological events, such as lung injury induced by a variety of contributing factors. Multiple lines of evidence suggest that the location of CD73, in proximity to adenosine receptor subtypes, indirectly determines its positive or negative effect in a variety of organs and tissues and that its action is affected by the transfer of nucleoside to subtype-specific adenosine receptors. Nonetheless, the bidirectional nature of CD73 as an emerging immune checkpoint in the pathogenesis of lung injury is still unknown. In this review, we explore the relationship between CD73 and the onset and progression of lung injury, highlighting the potential value of this molecule as a drug target for the treatment of pulmonary disease.

## 1. Introduction

In recent years, the enzyme 5′-nucleotidase (CD73) has become a “star molecule” in the study of immunology and tumor biology, as it controls the balance between ATP, ADP, AMP, and adenosine release in the extracellular space, modulating downstream inflammatory signaling transduction such as NLRP3 inflammasome and pyroptotic cell death [1]. CD73 was initially discovered in the 1930s and named 5′-nucleotidase (5′-NT) [2]. It was renamed ‘cluster of differentiation (CD) 73’ in 1989, when researchers identified a 69-kDa protein by immunoprecipitation [3]. Mechanistically, it is a crucial link of the purinergic system that participates in the regulation of cellular homeostasis, stress responses, tissue damage, and pathology in multiple organ systems [4]. Even more astonishingly, CD73 not only serves as an enzyme, but also as an adhesion molecule that facilitates the migration of normal and malignant cells [5]. Thus, the name CD73 is widely used in the recent literature (during the last decade), as interest in its immune functions in cancer has gradually grown.

CD73 is characterized structurally by an N-terminal domain containing zinc and cobalt ion-binding sites and a C-terminal domain containing the AMP-binding site [6]. A GPI anchor connects the C-terminal domain to the plasma membrane. In addition, a shortened soluble form of CD73 that retains ecto-5′-nucleotidase activity, which can be released from the plasma membrane and can circulate freely in the bloodstream, or other biological fluids, and exerts effects at a certain distance from the release site has also been described [7]. It has been proposed that soluble nucleotidases could be an important auxiliary effector system for the inactivation of acutely elevated nucleotides at sites of injury and inflammation [8]. Furthermore, hypoxia may induce soluble CD73 activity, which is necessary for hypoxia-induced plasma adenosine and oxygen release capability to prevent tissue hypoxia, inflammation, and lung damage [9].

CD73 is widely expressed in a variety of tissues, including the colon, kidney, brain, liver, heart, lung, spleen, and bone marrow [10]. CD73 accumulates in endothelial and epithelial cells, where it participates in the leukocyte trafficking program [11,12,13]. CD73 is also present on subsets of T and B cells that play a role in the immune system [14]. Extracellular ATP is a danger signal generated by dying or damaged cells, and it mediates inflammatory responses via P2 purinergic receptors (P2XRs and P2YRs). Several enzymes, collectively known as ectonucleotidases, rapidly convert extracellular ATP into adenosine. These include the ectonucleoside triphosphate diphosphohydrolase (ENTPD) and ectonucleotide pyrophosphatase/phosphodiesterase (ENPP) protein families [15]. Subsequently, extracellularly released ATP and ADP are degraded into AMP by ENTPD1 (CD39), followed by the hydrolysis of AMP to adenosine by CD73 [16]. Once released, adenosine induces signaling transduction via specific G-protein-coupled receptors (A1, A2A, A2B, A3), resulting in a variety of physiological responses including the regulation of oxygen supply/demand, inflammation, and other activities [17]. In addition to being dependent on the proximity of adenosine-sensitive P1 receptors, the amplitude of CD73 activity is also subject to the regulation of extracellular AMP-deviating routes because the CD38/CD203a/CD73 axis circumvents the canonical catabolic pathway mediated by the nucleoside tri- and diphosphohydrolase CD39 [18]. The function of CD73 is illustrated in Figure 1.

As the primary organ for gas–liquid exchange in mammals, the lung is an easy target for a wide variety of airborne pathogens, toxicants (aerosols), and allergens. CD73 is believed to contribute to the pathogenesis of lung injuries caused by a variety of stimuli. This review discusses the role of CD73 in lung injury and clarifies its molecular mechanisms, highlighting its potential value as a molecular target and biomarker in the treatment of pulmonary disease.

## 2. CD73 Exerts Bidirectional Modulatory Effects on Lung Injury

### 2.1. CD73 Enhances Lung Injury via Boosting Cell Invasion and Tumor Formation

Pathogens or other factors accelerate lung injury or pulmonary hypofunction, with pneumonia, acute respiratory distress syndrome (ARDS), and acute lung injury or inflammation (ALI) serving as representative examples. Recent studies have revealed CD73’s function as a biomarker for detecting lung injuries, making it an attractive therapeutic target for lung injuries induced by multiple inducers. CD73 knockdown significantly mediated airway epithelial cell wound repair through activation of the PKC alpha/MEK/ERK/p90RSK/CREB pathway [19]. The connection between CD73 and ARDS caused by the 2019 novel coronavirus disease (COVID-19) is also evident [20]. In the absence of treatment, lung cancer and the failure of pulmonary metastases are significant contributors to lung dysfunction and even pose a grave threat to life. The down-regulation of CD73 activity and expression exerts significant therapeutic effects in a variety of animal and cell models, according to a growing body of evidence [21,22]. Given that CD73 promotes tumor formation, invasion, and metastasis dysfunction [23,24], small-molecule inhibitors or mAb of CD73 have the potential to become essential targeting anticancer agents [25,26].

Based on current research, the most potent competitive CD73 inhibitor is α,β-methylene-ADP (APCP, adenosine-5′-O-[(phosphonomethyl)phosphonic acid]), which is capable of suppressing AMP hydrolysis in various cell lines and tissues at a low micromolar range [27], and a small molecule, AB680, has been used in clinical trials to treat pancreatic cancer. Although the combination of anti-CD73 antibodies with other therapies (e.g., anti-PD-1 mAbs) has shown promise as a therapy for carcinomas, further research is necessary [28,29,30]. The clinical application of anti-CD73 drugs is not yet fully successful. However, there are specific questions regarding the combination that merit careful consideration, including administration sequence, time window, dosage, etc.

Notably, CD73 is not always harmful because it promotes lung tissue-specific homeostasis, as will be discussed in greater detail below.

### 2.2. CD73 Maintains Lung Tissue-Specific Homeostasis and Respiratory Function

CD73 is a crucial endogenous enzyme of homeostasis that maintains the equilibrium between tissue inflammation and repair processes in a variety of pathological states, and prevents autoimmunity. CD73 is widely expressed on the surface of numerous lung cell types, including endothelial cells, airway epithelial cells, pneumocytes, and macrophages [31]. CD73 in epithelial cells was reported to contribute to ALI by decreasing endothelial permeability [32]. In addition to this essential function, the activity of CD73 on the mucosal and serosal surfaces of human airway epithelia is also advantageous to extracellular adenosine production [33]. Contrary to popular belief, extracellular adenosine regulates the movement of cilium and ion transport [34,35]. As depicted in Figure 2, CD73 on endothelial cells and airway epithelial cells protects against infectious and non-infectious lung diseases.

Another function of CD73 is to influence the neuronal respiratory drive to the diaphragm and the transmission between the phrenic nerve–diaphragm neuromuscular, and this can be achieved by reversing fatigue and promoting rehabilitation by enhancing ATP hydrolysis and reinforcing A2A facilitatory receptors’ co-localization, thereby facilitating the formation of adenosine [36].

### 2.3. CD73 Protects against Lung Injury in Hypoxia and Hyperoxia Condition

Alveolar–capillary exchange transports oxygen into the capillary vessels of the lung, and abnormal oxygen conditions may initiate multiple pulmonary diseases. ALI, marked by diffuse alveolar damage, is relevant to both hypoxic and hyperoxic environments. On the one hand, acute exposure to hypoxia alone can cause ALI-like phenotypes [37,38], whereas hypoxia-inducible factor (HIF) stabilization and HIF activator dimethyloxalylglycine treatment are protective against ALI during hypoxia [39,40,41]. On the other hand, excessive oxygen concentration (>0.21) can cause pulmonary damage and maximize ALI associated with mechanical ventilation due to an increase in reactive oxygen species (ROS) [42,43]. Hyperoxia, despite being a necessary therapy during profound hypoxemia in COVID-19, may have adverse long-term effects on clinically evident lung injury, adverse remodeling, and may result in parenchymal fibrosis.

Regarding hypoxia and hyperoxia, CD73 is a promising modulator for lung tissue homeostasis. In hypoxic *NT5E*^−/−^ mice, which are genetically predisposed to vascular leakage, administration of soluble CD73 can partially reverse the condition [44]. CD73 is protective against vascular leakage during hypoxic lung injury, which can be indirectly targeted via IFN-β-1α treatment [45,46]. Under conditions of hyperoxia, *NT5E*^−/−^ mice develop severe pulmonary edema, whereas WT mice exhibit elevated levels of CD73 [47]. Specifically, severe hyperoxia (95% O_2_) significantly harms the alveolar development of newborn mice, which is even more severe in *NT5E*^−/−^ mice. After eleven days of exposure, the mortality rate of *NT5E*^−/−^ mice was 100%, whereas the mortality rate of WT mice was 44%. Compared to a less severe environment (70% O_2_), *NT5E*^−/−^ mice had a higher lung infiltration of macrophages and lymphocytes than WT mice. Thus, CD73 protects against lung damage under specific oxygen conditions, as evidenced by adaptation to hypoxia and protection against hyperoxia.

### 2.4. CD73 in Inflammation-Driven Lung Injury: A Double-Edged Sword

As a vital immune organ, the lung contains both innate and adaptive immune cells that activate the immune system and induce inflammation via the release of a large quantity of cytokines in response to lung injury or infection. Accordingly, the physiologic inflammatory response is anticipated to contain invasive pathogens, heal the lung injury, and then restore homeostasis. However, an exaggerated inflammatory response is frequently the cause of excessive secondary lung injury. Due to inflammation driven by a complex set of mediators, we consider a spectrum of promising strategies for modifying inflammation in lung injury, such as controlling correlative inflammatory pathways (NLRP3 inflammasome), reducing the production of pro-inflammatory cytokines, and manipulating adaptive immunity.

It is well known that adenosine attenuates potentially harmful aspects of inflammation in various organs, such as pancreatitis, neuroinflammation, and pneumonia, in which CD73 is an essential component of the molecular signaling system [48,49]. As an illustration, ARDS, a typical uncontrolled inflammatory response, resulted in alveolar–capillary barrier damage and pulmonary vascular leakage. By affecting CD73 on epithelial and endothelial cells, IFN-β-1α administration prevented vascular leakage in animal models and inhibited leukocyte recruitment [50]. In contrast, in a randomized clinical trial, CD73 levels did not differ significantly between the placebo and IFN-β-1α groups [51]. Current research indicates that glucocorticoids inhibit the transduction of IFN-β-1α signaling, thereby inhibiting the upregulation of CD73, which may account for the negative outcome of the trial [52]. Anti-inflammatory properties of CD73-generated adenosine are commonly attributed to the benefits of pneumococcus infection promotion. Intriguingly, polymorphonuclear leukocytes (PMNs) mediate pulmonary inflammation in pneumonia infection by regulating bacterial load, which is closely associated with CD73 enzyme [53,54]. Although there was no discernible effect on PMN recruitment by suppression or reduction of CD73, within the first 6 h after intratracheal inoculation of mice, the number of PMN in the pulmonary interstitium increased significantly after 18 h of infection, reaching a peak three days after infection [55]. Recent studies indicate that the production of extracellular adenosine by CD73 inhibits the production of IL-10 by PMNs in response to pneumonia [56]. In fact, CD73 modifies the recruitment and bactericidal function of PMNs, thereby providing a therapeutic strategy for regulating potentially harmful inflammatory host responses during Gram-positive bacterial pneumonia. Similarly, CD73-derived adenosine is an important regulator of the inflammatory/immune response in animal models of allergic airway inflammation [57,58]. As demonstrated by the experimental data, both the upregulation of CD73 expression and activity in the lung of OVA-sensitized WT mice represent a self-protective mechanism that may regulate lung injury and tissue damage caused by inflammation [59,60]. Immediately following these findings, the levels of Th2 cytokines, IL-4, and IL-5 in the lungs of CD73 knock-out mice are significantly higher than those of OVA-sensitized WT mice [61]. In addition, treatment with *L. delbrueckii* UFV-H2b20 protected mice against airway inflammation, as indicated by a greater number of CD73+ Treg cells in the lungs [62]. CD73 may be a prognostic biomarker for allergic asthma, given the evidence presented above.

Interestingly, increased CD73 expression and activity in inflamed tissue coincide with persistent extracellular adenosine accumulation, which may promote pathological tissue remodeling, resulting in chronic inflammation and fibrosis with time [63,64,65]. Zhi Tian and colleagues used the long-term cigarette smoke model and discovered that CD73 was highly up-regulated, and adenosine production was stimulated, which led to the activation of A2B receptor, which exerted proinflammatory activity as evidenced by numerous studies [19,66]. Consequently, the role of CD73 in inflammation is not set in stone as the disease progresses, and it is crucial to analyze CD73’s function in order to develop therapeutic strategies for pulmonary inflammatory injury.

### 2.5. Blocking of CD73 Alleviates Carcinoma-Potentiated Lung Injury

Lung cancer is the type of malignant tumor with the highest mortality rate, accounting for 18% of all cancer deaths worldwide [67]. On the basis of the type of pathological differentiation, it is possible to distinguish between non-small cell lung cancer (NSCLC) and small cell lung cancer (SCLC) [68]. The former one accounts for 80–90% of lung cancers. Typical lung injury caused by cancer includes the alteration of bronchial secretion, obstructive pneumonia, and atelectasis. In addition, lung cancer induces a variety of immune responses that help eliminate tumor cells and inhibit their growth or, in some instances, aid tumor immune cells in evading detection.

In preclinical models, numerous studies have demonstrated that increased tumor growth and tumor immune evasion are associated with the expression and activity of the CD73/adenosine system in the tumor microenvironment (Table 1) [69,70]. Inhibition or deficiency of CD73, whether pharmacologic or genetic, promotes antitumor immunity by interfering with the conversion of extracellular ATP to adenosine [71]. The activation of A2A and A2B receptors is essential for CD73 to play an energetic role in an immunosuppressive environment, which is expressed on multiple immune cell types, including natural killer (NK) cells, effector, and regulatory T (Treg) cells [72,73,74]. In greater detail, the accumulation of CD73-generated adenosine activates the downstream A2A receptor, which reduces the cytotoxicity of CD8+ T cells and NK cells, and promotes the differentiation of CD4+ T cells into Treg cells [75].

Signaling via the A2B receptor accelerates the expansion and proliferation of myeloid-derived suppressor cells (MDSCs), and CD73 activity on MDSCs inhibits T cells and NK cells [72]. Numerous experiments conducted on EGFR-mutant NSCLC cell lines have conclusively demonstrated that long-term alterations in the activation of purinergic signaling pathways can promote the development of metabolic perturbations and chronic inflammation. The overexpression of CD73 has no effect on the proliferation of CD4+ and CD8+ T cells in EGFR-mutant individuals, but it is closely associated with the elevation of CD4+ FoxP3+ Treg cells [78,101]. Administration of anti-CD73 regent increases tumor cell death in vitro, and CD73 blockade exhibits slower tumor outgrowth and significantly increased antigen-specific CD8+ T-cell immunogenicity following treatment with pemetrexed in an EGFR-mutant lung cancer mouse model [30,102]. Another aspect that is worth mentioning is the lung metastases of multiple types of cancer. A recent study found that inhibiting CD73 activity and expression could not only inhibit cell migration and invasion in human triple-negative breast cancer and mouse 4T1 cell lines, but also significantly suppress lung metastasis of 4T1 cells in a xenograft animal model [88]. Consistently, additional studies have demonstrated that CD73 overexpression facilitates migration and invasion in other types of cancer, such as gastric cancer and ovarian cancer, etc. [103,104]. Notably, the relationship between CD73 and epithelial-to-mesenchymal transition is a central aspect of cancer metastasis, and preliminary findings suggest that CD73 may serve as a prognostic marker due to its role in promoting epithelial to mesenchymal transition progression, particularly in lung adenocarcinoma and triple-negative breast cancer [89,92]. It should not be forgotten that lung-related complications are inevitable with cancer treatment, specifically radiation-induced lung fibrosis. Radiation-induced alterations promote alternative polarization of recruited immune cells, pathologic immune cell impact, and excessive tissue remodeling, which sustain repair processes and chronic inflammation, thereby facilitating tissue scarring, fibrosis, and secondary tumor formation [105,106,107]. The exposure of WT mice to a single dose (15 Gray) of whole-thorax irradiation induces the increased activity of CD73 in the lung for 3 to 30 weeks and lung fibrosis for 25 weeks, whereas CD73-deficient mice exhibited significantly less radiation-induced lung fibrosis (*p* < 0.010). In addition, treatment with CD73 antibodies significantly reduces radiation-induced lung fibrosis in WT mice [108]. Figure 3 depicts the major biomarkers and pathways in which CD73 participates in influencing lung injury.

## 3. Discussion

Lung injury is a common and frequent disorder that still represents critical unmet clinical needs. It is the undesirable result of a variety of factors, including abnormal oxygen environment, pathogens, uncontrollable inflammatory response, cancer, and improper treatment. Current therapeutic strategies are primarily focused on three objectives: (1) restoring the damaged pulmonary function of ventilation and air exchange; (2) halting the progression of underlying diseases; and (3) preserving lung homeostasis. When the effects of a lung injury are irreversible, these proposals are inadequate. As stated in the previous chapters of this review, CD73 plays multiple roles in the initiation and progression of lung injury, and is a key regulator in the purinergic signaling pathways. Considering the species differences in CD73 regulation and related disease types, such as the contradiction between the preclinical findings in mice and humans induced by hyperoxia [4], it will be crucial for in-depth studies to analyze results from *NT5E*^−/−^ mice and human-derived models, including primary tissues, pluripotent stem cells (iPSCs), and tissue organoids [109,110]. The anti-inflammatory property of CD73 in lung injury is related to adenosine, whereas a prolonged increase in adenosine production can promote pulmonary inflammation and airway remodeling, resulting in pulmonary tissue destruction [64]. Therefore, it is essential to assess the extent of adenosine elevation in this issue. Changes in CD73 expression may also affect the ratio of extracellular ATP to adenosine, which may result in a radical alteration of the lung microenvironment. Assessing how CD73 balances ATP, ADP, and adenosine is the subject of ongoing research.

Although many of the studies cited in this article demonstrate that CD73 is variable in lung injury induced by various stimuli, only a few describe the variation of CD73 at all biological levels, including mRNA, protein expression, enzymatic activity, etc. For one, there is not always a correlation between increased protein expression and increased enzymatic activity, and it is possible that mRNA upregulation does not result in increased protein expression. Moreover, it is impossible to ignore the roles played by protein modifications such as phosphorylation and ubiquitination. Consequently, one of the focuses of future research into CD73 regulation will be the entire spectrum of multi-level changes. In animal experiments, numerous additional research tools, such as radiolabeled antibodies and fluorescent probes [111,112], have been utilized to monitor CD73 expression and localization. Numerous bottlenecks will inevitably be overcome with the help of tools, and these tools provide the best possible opportunities to comprehend CD73 biology and regulation during lung injury.

Despite the recent emergence of novel effective and selective medications and/or inhibitory antibodies against CD73 [27], there are many uncertainties or even contradictory data regarding the pathogenic versus the beneficial role of this protein in the lung. In fact, the proximity of CD73 to subtype-specific adenosine receptors indirectly determines whether its effect is positive or negative, and its action is influenced by the transfer of nucleoside to subtype-specific adenosine receptors. As for CD73’s double-edged sword property, it participates in the inflammatory response via the A2A receptor, followed by the binding to adenosine, which prevents the activation and proliferation of CD4+ T cells, and simultaneously inhibits the immune response of Th1 cells and Th17 cells in vivo and promotes Treg, thereby exerting an anti-inflammatory effect under conditions of general inflammation-driven lung injury. In other conditions characterized by chronic, persistent inflammation, such as cancer, this is not the case. On the one hand, CD73 also plays an anti-inflammatory role, which not only reduces the release of pro-inflammatory mediators but also inhibits the attack of immune cells on tumor cells. In contrast, CD73 promotes the growth of tumor cells by activating the A2B receptor and facilitating the secretion of tumor growth-promoting factors such as interleukin (IL)-6, IL-8, transforming growth factor (TGF)-beta, etc. [113]. In addition, it is important to recognize the strong connection between CD73 and A2A receptors in a variety of tissues, such as the brain, colon, neuromuscular junction, etc., and to rigorously correct this functional cross-talk between different tissues. Enhancement of the CD73/A2A adenosine metabolic pathway, for example, alleviated ulcerative colitis induced by DSS [114]. CD73 suppression and adenosine deaminase upregulation rescued post-inflammatory ileitis through neuromodulation, as adenosine mediated both anti-inflammatory (via A2A) and pro-inflammatory (via A1 and A2B) activities [115]. CD73 generates an auto-regulatory feed-forward adenosine formation to activate A2A receptor and promote neuroinflammation [116]. Moreover, CD73 is not considered an adhesion molecule that specifically targets immunocyte cells; consequently, it has no signaling capacity unless it regulates adenosine formation, which is capable of mediating a bidirectional inflammation-modulatory effect through adenosine-A1/A2B receptors for pro-inflammation and adenosine-A2A receptors for anti-inflammation. Consequently, it is necessary to differentiate the effect of CD73 during lung injury in various circumstances.

In addition, a variety of other aspects of CD73’s crucial role in maintaining the homeostasis of other systems have been addressed. CD73 is one of the key molecules involved in body homeostatic processes, including the control of epithelial barrier function and the regulation of secretive and/or reabsorptive processes at the intestinal level [117]. Although animal models lacking CD73 or treated with anti-CD73 drugs do not demonstrate significant adverse effects, these potential adverse effects should be carefully considered before this therapeutic strategy is implemented in clinical practice. Second, the administration of an anti-CD73 agent may induce a supraphysiological immune response, which may disrupt the immune system’s equilibrium and result in autoimmune disorders. In addition, the variability of local tumor microenvironments may have a significant impact on the response to anti-CD73 therapy. Thus, the time is ripe for the development of new small molecules with an inhibitory effect on CD73, as the clinical application is anticipated in the near future.

In conclusion, the significance of CD73 is particularly evident in the purinergic signaling pathway and its involvement in the development of lung injury has significant value and dual nature. There is accumulating evidence that targeting CD73 has a positive effect on lung injury, but the field of tumor immunotherapy is a hotspot and is making critical progress. AB680 has been used clinically to treat pancreatic cancer. Although lung injury therapy targeting CD73 has not yet been scheduled for clinical use [118], the effect of CD73 in preclinical models suggests that CD73 may be a valuable target of pulmonary disease in the future.

## Figures and Tables

**Figure 1 ijms-24-05545-f001:**
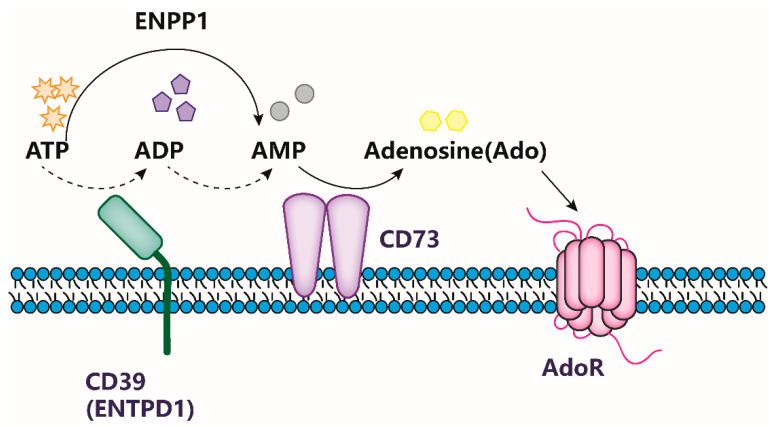
CD73 accelerates the accumulation of ADO. Purine metabolism by the ecto-nucleotidase CD73 is widespread. CD73, a plasma membrane GPI-anchored glycoprotein, collaborates with CD39 to convert ATP to AMP. Ecto-nucleotide pyrophosphatase/phosphodiesterase-1 (ENPP1) can directly convert ATP to AMP. CD73 is the main enzyme that dephosphorylates AMP to produce extracellular adenosine (Ado); however, tissue non-specific alkaline phosphatase (TNAP) or prostatic acid phosphatase (PAP) can also carry out this activity.

**Figure 2 ijms-24-05545-f002:**
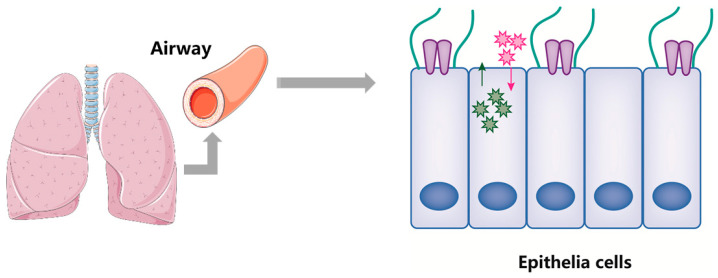
CD73 supports lung tissue-specific homeostasis in epithelial cells to control respiratory functions. CD73 is the primary source of extracellular adenosine on airway epithelial surfaces in the respiratory system. Mucociliary clearance and ion exchange, such as chloride, are both regulated by extracellular adenosine, which also helps to maintain the integrity of the tissue barrier by lowering endothelial permeability. Red and black arrows represent intracellular and extracellular ion exchange.

**Figure 3 ijms-24-05545-f003:**
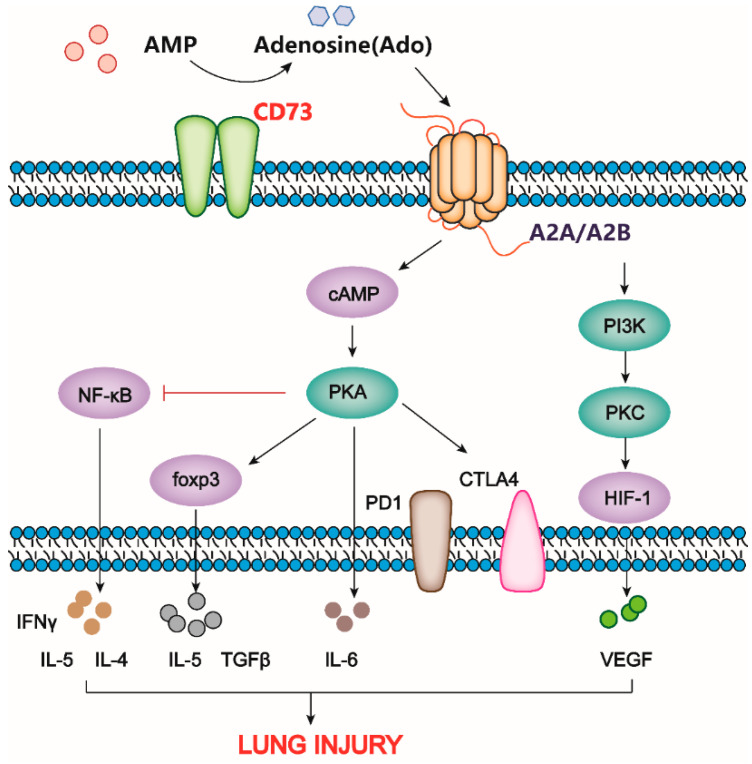
CD73 modulates downstream signaling pathways to affect lung injury. CD73 mediates lung injury via activating adenosine receptors A2A and A2B. They often suppress the function of a variety of immune cells by activating cyclic AMP (cAMP)-dependent PKA and inhibiting the nuclear factor-B (NF-B) signaling pathways. In addition to downstream cytokines, the up-regulation of CD73 also enhances the expression of co-inhibitory receptors cytotoxic T lymphocyte antigen 4 (CTLA4) and PD1. In response to hypoxia, CD73-mediated A2A receptor activation can regulate the PIK3 signaling pathway.

**Table 1 ijms-24-05545-t001:** Role of CD73 in different types of lung injury.

Type of Lung Injury	Classification	CD73			Reference
		Expression	Relevant Mechanism	Positive (P) or Negative (N)	
Cancer	NSCLC	↑	LDH5 and HIF1α↑, PD-L1 and LDHA↑	N	[76]
		↓	IL-10↓, CXCR4↓	P	[77]
		↓	CAF/T cell interactions	P	[78]
	EGFR-mutated NSCLC	↓	CD8+ T cells↑, IFN-γ, and TNF-α↑	P	[30]
		↑	PD-L1↑	N	[79]
	LUAD	↑	PD-L1↑	N	[80]
		Marker	-	-	[81]
	SCLC	↓	Elimination of metastatic chemoresistant SCLC	P	[82]
		Marker	-	-	[83]
	NK cell	↑	LAG-3, VISTA, PD-1, and PD-L1↑,CD4-positive T cell, and IFN-γ↓	N	[84]
		↓	TME↓	P	[85]
	I–III LUSC	Marker	-	-	[86]
Pulmonary metastases	BC	↓	ADO-activated intracellular A2A receptor signaling pathway is linked to the AKT-β catenin pathway to regulate BC cell invasiveness and metastasis to the lung		[87]
	Triple-negative BC	↓	LC3I/LC3II ratio and p62↑	P	[88]
		↓	TNBC cell migration in both normoxia and hypoxia↓	P	[89]
	LuM-1	↓	IFN-γ and cytotoxicity against LuM-1↑	P	[90]
	Tumor-bearing mice	↓	A2B↓	P	[91]
	LUAD	↑	Epithelial to mesenchymal transition (EMT) progression↑	N	[92]
Bacterial infection	MBCs	Marker	Coexpression of at least 2 of these 3 memory markers distinguishes MBCs likely to differentiate into antibody-secreting cells (ASCs) upon reactivation	-	[93]
	PMNs	↓	IL-10↑	N	[56]
ARDS	EVs with MSC origin	Marker	-	-	[94]
	Lung organ cultures	↓	IFN beta-1a signaling↓	N	[52]
Hypoxic stress	NCI-H292	↓	apoptosis↑, sensitivity to mitomycin↓, sensitivity to vincristine↑	P	[95]
Allergic airway inflammation	OVA	↓	IL-4 and IL-5↑, TGF-β↓, CD4+CD25+Foxp3+ T cells↓	p	[61]
	OVA	↓	CD23+ B cells and IL4+ T cells↓, mast cells and degranulation↓	P	[96]
Long-term cigarette smoke	Cigarette smoke	↓	Inflammatory cells↓, IL-6↓	P	[19]
COPD	COPD	Marker	-	-	[97]
HIV	DN T-cells	Marker	-	-	[98]
Thoracic endometriosis	TH-EM1 cell	Marker	-	-	[99]
Tuberculosis	MP287/03-infected mice	Marker	-	-	[100]

## Data Availability

Not applicable.

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
