# Peer review of "CD73: Friend or Foe in Lung Injury"

_ijms, 2023, doi:10.3390/ijms24065545_

Round 1

Reviewer 1 Report

Manuscript ID: ijms-2216039 - CD73: Friend or Foe in Lung Injury

In this paper, Hu et al. critically revised the recent literature on the pros and cons of targeting CD73/ecto-5´-nucleotidase for the treatment of lung diseases.

Although the topic is appealing given that new highly potent and selective drugs and/or inhibitory antibodies against CD73 have recently emerged, there are too many uncertainties (and even contradictory facts) regarding the pathophysiological vs the protective role of this molecule in the lung. In addition to these complexities, many other aspects regarding the critical role of CD73 in other systems’ homeostasis have not been adequately accounted for in this review, but may be sufficiently relevant to prevent the usage of the proposed therapeutic alternative. These limitations should be briefly discussed.

This review misses completely the relevance of CD73 in the control of respiration, which is paramount to lung (patho)physiology. CD73 affects both the neuronal respiratory drive to the diaphragm (the more important respiratory muscle) and the phrenic nerve-diaphragm neuromuscular transmission by preventing fatigue and facilitating rehabilitation by favouring adenosine formation from the breakdown of released ATP and activation of co-localized A2A facilitatory receptors.

Authors should also clearly emphasize that the positive and negative effects of CD73 indirectly rely on the localization of this enzyme in the proximity of adenosine receptor subtypes and its actions depend on the delivery of the nucleoside to subtype-specific adenosine receptors. The close relationship between CD73 and A2A receptors in several tissues (e.g. brain, intestine, neuromuscular junction) should be acknowledged and this functional crosstalk critically revised. In fact, on its own CD73 has no signalling ability if one discounts its role as an adhesion molecule directly targeting immunocyte cells.  

Besides dependence on the proximity of adenosine-sensitive P1 receptors, the magnitude of CD73 actions is also subject to the control of extracellular AMP-deviating pathways, like that provided by AMP deaminase bypassing adenosine formation.

Other issues:

1.      Abstract – The first sentence overstates the real importance of CD73 by assuming actions to this molecule that are beyond its ability as a cell marker/ecto-nucleotidase enzyme. This must be attenuated. The abstract should be fully rewritten since it is too vague and not compatible with the standards of a scientific paper. It should start by stating the major molecular properties of CD73 for a lay audience. Then, it should objectively focus on the main controversies motivating authors’ interest in this topic.   

2.      Authors should revise the entire MS as they seem to use without distinction adenosine “release as such” (from equilibrative nucleoside transporters) and “release of adenosine” from the extracellular ATP by the action of CD73. The latter term is not corrected (should be replaced by “formed” or “generated”).

3.      The molecular characteristics of CD73 should be carefully stated. CD73 is a GPI-anchored protein that can be cleaved from the plasma membrane (soluble forms) and thereby may exert effects at a certain distance from the release site. Implications of this phenomenon in the lung should be addressed. The C- and N-terminals should be located either intra- or extracellularly.

4.      Countless misspelling errors and unacceptable grammatical issues are perturbing the readability of the MS. The paper must be fully revised by a native English-speaking person.  For instance, “mechanically” perhaps should be “mechanistically” (page 1, line 36).

5.      Figure legends are too telegraphic. Should be rewritten for missing aspects (e.g. Figure 2).

6.      Table 1 should be reorganized by specific pathological conditions and/or by pros- and cons- of CD73 inhibition to increase its clinical impact. This reorganization will also emphasize controversies about the role of CD73 under similar pathological conditions.

Author Response

Dear Editor

On behalf of my co-authors, I thank you very much for your great support and the reviewers’ comments concerning our manuscript entitled “CD73: Friend or Foe in Lung Injury” (manuscript ID:ijms-2216039). We have read through comments carefully and have made corrections. We have tried our best to improve our revised manuscript following the reviewer’s wonderful comments and recommendations. Hope it will be acceptable for considerations to publication at International Journal of Molecular Science. If you have any additional questions or suggestions, please let us know. Thanks again in advance.

Best wishes,

Jiasi Wu

Ph.D, Lecturer

Chengdu University of Traditional Chinese Medicine,

Chengdu, 611137, P.R. China

PS: Point-by-point response to reviewer’s comments. All corrections in the text were highlighted in yellow.

Reviewer #1:

Q1: Although the topic is appealing given that new highly potent and selective drugs and/or inhibitory antibodies against CD73 have recently emerged, there are too many uncertainties (and even contradictory facts) regarding the pathophysiological vs the protective role of this molecule in the lung. In addition to these complexities, many other aspects regarding the critical role of CD73 in other systems’ homeostasis have not been adequately accounted for in this review, but may be sufficiently relevant to prevent the usage of the proposed therapeutic alternative. These limitations should be briefly discussed.

A: These limitations mentioned above have been briefly discussed(Page 11, line 352). Details are as follows,

Despite the recent emergence of novel effective and selective medications and/or inhibitory antibodies against CD73, there are many uncertainties or even contradictory data regarding the pathogenic versus the beneficial role of this protein in the lung.

Q2: This review misses completely the relevance of CD73 in the control of respiration, which is paramount to lung (patho)physiology. CD73 affects both the neuronal respiratory drive to the diaphragm (the more important respiratory muscle) and the phrenic nerve-diaphragm neuromuscular transmission by preventing fatigue and facilitating rehabilitation by favouring adenosine formation from the breakdown of released ATP and activation of co-localized A2A facilitatory receptors.

A: It has been added in the text at Page 4 line 134. Details are as follows,

Another function of CD73 is to influence the neuronal respiratory drive to the diaphragm and the transmission between the phrenic nerve-diaphragm neuromuscular, and this can be achieved by reversing fatigue and promoting rehabilitation by enhancing ATP hydrolysis and reinforcing A2A facilitatory re-ceptors co-localization, thereby facilitating the formation of adenosine (Oliveira L, Correia A, Cristina Costa A, et al. Deficits in endogenous adenosine formation by ecto-5'-nucleotidase/CD73 impair neuromuscular transmission and immune competence in experimental autoimmune myasthenia gravis. Mediators Inflamm. 2015;2015:460610. doi:10.1155/2015/460610).

Q3: Authors should also clearly emphasize that the positive and negative effects of CD73 indirectly rely on the localization of this enzyme in the proximity of adenosine receptor subtypes and its actions depend on the delivery of the nucleoside to subtype-specific adenosine receptors. The close relationship between CD73 and A2A receptors in several tissues (e.g. brain, intestine, neuromuscular junction) should be acknowledged and this functional crosstalk critically revised. In fact, on its own CD73 has no signalling ability if one discounts its role as an adhesion molecule directly targeting immunocyte cells.

A: It has been added in the text at Page 12 line 370. Details are as follows,

In addition, it is important to recognize the strong connection between CD73 and A2A receptors in a variety of tissues, such as the brain, colon, neuromuscular junction, etc., and to rigorously correct this functional cross-talk between different tissues. Enhancement of the CD73/A2A adenosine metabolic pathway, for example, alleviated ulcerative colitis induced by DSS. CD73 suppression and adenosine deaminase upregulation rescued post-inflammatory ileitis through neuromodulation, as adenosine mediated both anti-inflammatory (via A2A) and pro-inflammatory (via A1 and A2B) activities. CD73 generates an auto-regulatory feed-forward adenosine formation to activate A2AR and promote neuroinflammation. Moreover, CD73 is not considered an adhesion molecule that specifically targets immunocyte cells; consequently, it has no signaling capacity unless it regulates adenosine formation, which is capable of mediating a bidirectional inflammation-modulatory effect through adenosine-A1/A2B receptors for pro-inflammation and adenosine-A2A receptors for anti-inflammation(Zhu Y, Zhuang Z, Wu Q, et al. CD39/CD73/A2a Adenosine Metabolic Pathway: Targets for Moxibustion in Treating DSS-Induced Ulcerative Colitis. Am J Chin Med. 2021;49(3):661-676. doi:10.1142/S0192415X21500300;Vieira C, Magalhães-Cardoso MT, Ferreirinha F, et al. Feed-forward inhibition of CD73 and upregulation of adenosine deaminase contribute to the loss of adenosine neuromodulation in postinflammatory ileitis. Mediators Inflamm. 2014;2014:254640. doi:10.1155/2014/254640;Meng F, Guo Z, Hu Y, et al. CD73-derived adenosine controls inflammation and neurodegeneration by modulating dopamine signalling. Brain. 2019;142(3):700-718. doi:10.1093/brain/awy351).

Q4: Besides dependence on the proximity of adenosine-sensitive P1 receptors, the magnitude of CD73 actions is also subject to the control of extracellular AMP-deviating pathways, like that provided by AMP deaminase bypassing adenosine formation.

A: It has been added in the text at Page 2 line 68. Details are as follows,

In addition to being dependent on the proximity of adenosine-sensitive P1 receptors, the amplitude of CD73 activity is also subject to the regulation of extracellular AMP-deviating routes because the CD38/CD203a/CD73 axis circumvents the canonical catabolic pathway mediated by the nucleoside tri- and diphosphohydrolase CD39(Horenstein AL, Chillemi A, Zaccarello G, et al. A CD38/CD203a/CD73 ectoenzymatic pathway independent of CD39 drives a novel adenosinergic loop in human T lymphocytes. Oncoimmunology. 2013;2(9):e26246. doi:10.4161/onci.26246).

Q5: Abstract – The first sentence overstates the real importance of CD73 by assuming actions to this molecule that are beyond its ability as a cell marker/ecto-nucleotidase enzyme. This must be attenuated. The abstract should be fully rewritten since it is too vague and not compatible with the standards of a scientific paper. It should start by stating the major molecular properties of CD73 for a lay audience. Then, it should objectively focus on the main controversies motivating authors’ interest in this topic

A: We have made the necessary revisions to the abstract to more closely match the full text. Details are as follows,

CD73 (ecto-5'-nucleotidase) plays a strategic role in calibrating the magnitude and chemical nature of purinergic signals that are delivered to immune cells. Its primary function is to convert extra-cellular ATP to adenosine in concert with CD39 in normal tissues to limit excessive immune re-sponse in many pathophysiological events, such as lung injury induced by a variety of contributing factors. Multiple lines of evidence suggest that the location of CD73 in proximity to adenosine re-ceptor subtypes indirectly determines its positive or negative effect in a variety of organs and tissues and that its action is affected by the transfer of nucleoside to subtype-specific adenosine receptors. Nonetheless, the bidirectional nature of CD73 as an emerging immune checkpoint in the patho-genesis of lung injury is still unknown. In this review, we explore the relationship between CD73 and the onset and progression of lung injury, highlighting the potential value of this molecule as a drug target for the treatment of pulmonary disease.

Q6: Authors should revise the entire MS as they seem to use without distinction adenosine “release as such” (from equilibrative nucleoside transporters) and “release of adenosine” from the extracellular ATP by the action of CD73. The latter term is not corrected (should be replaced by “formed” or “generated”).

A: We have modified the terminology of the two classes of adenosine throughout the text.

Q7: The molecular characteristics of CD73 should be carefully stated. CD73 is a GPI-anchored protein that can be cleaved from the plasma membrane (soluble forms) and thereby may exert effects at a certain distance from the release site. Implications of this phenomenon in the lung should be addressed. The C- and N-terminals should be located either intra- or extracellularly.

A: It has been added in the text at Page 2 line 38. Details are as follows,

CD73 is characterized structurally by an N-terminal domain containing zinc and cobalt ion-binding sites and a C-terminal domain containing the AMP-binding site. A GPI anchor connects the C-terminal domain to the plasma membrane. In addition, a shortened soluble form of CD73 that retains ecto-5′-nucleotidase activity, which can be released from the plasma membrane and circulate freely in the bloodstream or other biological fluids, thereby exerting effects at a certain distance from the release site, has also been described. It has been proposed that soluble nucleotidases could be an important auxiliary effector system for the inactivation of acutely elevated nucleotides at sites of injury and inflammation. Fur-thermore, hypoxia may induce soluble CD73 activity, which is necessary for hypoxia-induced plasma adenosine and oxygen release capability to prevent tissue hypoxia, inflammation, and lung damage (Heuts DP, Weissenborn MJ, Olkhov RV, et al. Crystal structure of a soluble form of human CD73 with ecto-5'-nucleotidase activity. Chembiochem. 2012;13(16):2384-2391. doi:10.1002/cbic.201200426;Yegutkin GG. Nucleotide- and nucleoside-converting ectoenzymes: Important modulators of purinergic signalling cascade. Biochim Biophys Acta. 2008;1783(5):673-694. doi:10.1016/j.bbamcr.2008.01.024;Liu H, Zhang Y, Wu H, et al. Beneficial Role of Erythrocyte Adenosine A2B Receptor-Mediated AMP-Activated Protein Kinase Activation in High-Altitude Hypoxia. Circulation. 2016;134(5):405-421. doi:10.1161/CIRCULATIONAHA.116.021311).

Q8: Countless misspelling errors and unacceptable grammatical issues are perturbing the readability of the MS. The paper must be fully revised by a native English-speaking person.  For instance, “mechanically” perhaps should be “mechanistically” (page 1, line 36).

A: We have tried our best to correct the ubiquitous grammatical errors and modified the word you pointed out (page 1, line 30).

Q9: Figure legends are too telegraphic. Should be rewritten for missing aspects (e.g. Figure 2).

A: The figure legend has been tweaked and detailed instructions added. Fig 1 is on page 3. Fig 2 is on page 5. Fig 3 is on page 10. The details are displayed in the attachment.

Q10: Table 1 should be reorganized by specific pathological conditions and/or by pros- and cons- of CD73 inhibition to increase its clinical impact. This reorganization will also emphasize controversies about the role of CD73 under similar pathological conditions.

A: Table 1 has been reorganized by specific pathological conditions. In addition, the evaluation of the role of CD73 in the literature is added in the table. Table 1 is on page 7.The details are displayed in the attachment.

Reviewer 2 Report

The manuscript by Hu and coworkers tried to review the current understanding of the two-faced role of CD73 enzyme in lung injury. The review is easy to comprehend, well written and organized, table is very clear and the references are adequate. However, I have some comments:

- In Figure 1 authors should add the representation of the different adenosine receptors

- In table 1 CD73 in different type of lung injury, authors should add to allergic Airway Inflammation the model of allergic sensitization and challenge and the related results (Caiazzo E, Cerqua I, Turiello R, Riemma MA, De Palma G, Ialenti A, Roviezzo F, Morello S, Cicala C. Lack of Ecto-5'-Nucleotidase Protects Sensitized Mice against Allergen Challenge. Biomolecules. 2022 May 13;12(5):697. doi: 10.3390/biom12050697).

Author Response

Dear Editor

On behalf of my co-authors, I thank you very much for your great support and the reviewers’ comments concerning our manuscript entitled “CD73: Friend or Foe in Lung Injury” (manuscript ID:ijms-2216039). We have read through comments carefully and have made corrections. We have tried our best to improve our revised manuscript following the reviewer’s wonderful comments and recommendations. Hope it will be acceptable for considerations to publication at International Journal of Molecular Science. If you have any additional questions or suggestions, please let us know. Thanks again in advance.

Best wishes,

Jiasi Wu

Ph.D, Lecturer

Chengdu University of Traditional Chinese Medicine,

Chengdu, 611137, P.R. China

PS: Point-by-point response to reviewer’s comments. All corrections in the text were highlighted in yellow.

Reviewer #2:

Q1: In Figure 1 authors should add the representation of the different adenosine receptors

A: We have not subdivided the ADO receptor in figure 1, and the key ADO receptor is reflected in figure 3.The details are displayed in the attachment.

Q2: In table 1 CD73 in different type of lung injury, authors should add to allergic Airway Inflammation the model of allergic sensitization and challenge and the related results (Caiazzo E, Cerqua I, Turiello R, Riemma MA, De Palma G, Ialenti A, Roviezzo F, Morello S, Cicala C. Lack of Ecto-5'-Nucleotidase Protects Sensitized Mice against Allergen Challenge. Biomolecules. 2022 May 13;12(5):697. doi: 10.3390/biom12050697).

A: Table 1 has been reorganized and included this reference (page 8). (Caiazzo E, Cerqua I, Turiello R, et al. Lack of Ecto-5'-Nucleotidase Protects Sensitized Mice against Allergen Challenge. Biomolecules. 2022;12(5):697. Published 2022 May 13. doi:10.3390/biom12050697).The details are displayed in the attachment.
